# An Innovative Polarisation-Insensitive Perfect Metamaterial Absorber with an Octagonal-Shaped Resonator for Energy Harvesting at Visible Spectra

**DOI:** 10.3390/nano13121882

**Published:** 2023-06-19

**Authors:** Mohammad Jakir Hossain, Md. Habibur Rahman, Mohammad Rashed Iqbal Faruque

**Affiliations:** 1Department of Electrical and Electronic Engineering, Dhaka University of Engineering & Technology (DUET), Gazipur 1707, Bangladesh; habibur1310@gmail.com; 2Space Science Centre (ANGKASA), Universiti Kebangsaan Malaysia, Bangi 43600 UKM, Malaysia

**Keywords:** perfect metamaterial absorber, polarisation-insensitive, incident angle stability, solar energy harvesting, visible frequencies

## Abstract

Perfect metamaterial absorber (PMA) is an attractive optical wavelength absorber with potential solar energy and photovoltaic applications. Perfect metamaterials used as solar cells can improve efficiency by amplifying incident solar waves on the PMA. This study aims to assess a wide-band octagonal PMA for a visible wavelength spectrum. The proposed PMA consists of three layers: nickel, silicon dioxide, and nickel. Based on the simulations, polarisation-insensitive absorption transverse electric (TE) and transverse magnetic (TM) modes were achieved due to symmetry. The proposed PMA structure was subjected to computational simulation using a FIT-based CST simulator. The design structure was again confirmed using FEM-based HFSS to maintain pattern integrity and absorption analysis. The absorption rates of the absorber were estimated at 99.987% and 99.997% for 549.20 THz and 653.2 THz, respectively. The results indicated that the PMA could achieve high absorption peaks in TE and TM modes despite being insensitive to polarisation and the incident angle. Electric field and magnetic field analyses were performed to understand the absorption of the PMA for solar energy harvesting. In conclusion, the PMA possesses outstanding visible frequency absorption, making it a promising option.

## 1. Introduction

The escalating demand for energy for our daily use became apparent in 2013 when the aggregated global population consumed a staggering 15 terawatts (TW) of energy. There is a projected twofold increase in this statistic by the year 2050. The maintenance of the current levels of CO_2_ emission necessitates the acquisition of an additional quantum of renewable energy in the range of 13–15 Terawatts (TW) by 2050 [1]. Solar energy is regarded as an exceptional renewable energy resource for generating electricity, which primarily utilises photovoltaic (PV) cells. The amount of energy transmitted by electromagnetic (EM) radiation within a single hour can satisfy the annual energy requirements of the world. Luminous energy from solar radiation on Earth consists of approximately 7.5% of ultraviolet, 48% of visible, and 43% of infrared radiation. Consequently, researchers are focusing on generating solar absorbers with superior efficiency due to the ease of manufacturing.

Metamaterial absorbers are a viable option for wide-band solar absorption due to their unique properties. Metamaterial is an artificially designed structure exhibiting unique EM characteristics characterised by negative permittivity and permeability. The concept of metamaterial was initially proposed by Veselago in 1968. However, metamaterials are used considerably in engineering, particularly in perfect lensing, optical cloaking, antenna design, sensing technology, and absorber structures [2]. The use of metasurfaces for absorptive purposes represents a promising field of application for metamaterials. The concept of a metamaterial absorber (MMA) was initially introduced by Landy et al. in 2008 [3], wherein they fabricated an engineered MMA with utmost precision. Subsequently, a diverse range of metamaterials (MMAs) were incorporated across a multitude of EM frequency ranges, including ultra-narrow-band absorbers [4], broadband absorbers [5], refractory metamaterial absorbers [6], polarisation-insensitive configurations [7], multi-layer absorbers [8], ultra-broadband 3D structure absorbers [9], and larger-scale flat antennae [10]. An MMA is characterised by a triple architecture comprising layers of metal–insulator–metal (MIM). The metal layer at the uppermost position is called the resonator. The function of the metallic base layer of the structure is to terminate the EM waves. Meanwhile, the introduction of a dielectric layer between the metal film and the resonator plate facilitates the generation of coupling capacitance within the structure. The minimum reflection of the incident wave occurs when there is adequate alignment between the meta-surface impedance and the open space impedance [11]. For instance, Li et al. [12] presented a brief and adaptable perfect absorber featuring a strontium titanate (STO) crystal substrate, where the absorber was tuneable in dual-band. The design structure demonstrated a maximal absorption peak, reaching 97.97% and 95.92% at frequencies of 0.15 THz and 0.30 THz, respectively. On the other hand, Mulla et al. proposed the utilisation of a metamaterial-based absorber as a means of harvesting solar energy [13]. The proposed absorber in the study exhibited an efficiency of 98.2% at a frequency of 445.85 THz while achieving a higher absorption efficiency of 99.4% within the spectral range of 624 to 658.3 THz. The study employed aluminium (Al) and silicon dioxide (SiO_2_) as the metallic components of the construction and dielectric substance, respectively.

In another study, MPA-facilitated solar energy extraction was explored to leverage the entire visible light spectrum, as evidenced by recent studies [14]. Uniform absorption properties can be attained by adequately engineering electric and magnetic resonance. A plethora of research demonstrated varying configurations of PMA with restricted absorption characteristics, namely constrained frequency range and diminished absorption levels. The observable range of wavelengths was confined to the interval of 380–750 nm, which can be attributed to a frequency domain within the bounds of 400–790 THz. Incorporating polarisation-insensitive properties assumes fundamental significance in absorbent material design, owing to the distinctive absorption rate manifested across various incident polarisation angles [15]. An optical region absorber was constructed in the study from a tri-layered composition of tungsten and silicon dioxide, where the tungsten acted as a metamaterial [16].

The absorber exhibited an absorption bandwidth exceeding 91.24% within the wavelength range of 389.34 to 697.19 nm and attained a maximum absorption of 99.99%. Meanwhile, another study formulated a four-layer optical wavelength absorber composed of Cu, Si_3_N_4_, and silicon (Si) to operate within the 400–700 nm range [17]. The recorded minimum and peak absorption levels were approximately 80% and 97%, respectively. Meanwhile, Liu et al. demonstrated an absorption exceeding 83% in the visible wavelength region from 370 to 880 nm [18]. The utilisation of gold (Au) and silicon dioxide (SiO_2_) yielded a maximum absorption value of 92%. Another study reported a compact Ag and SiO_2_-based absorber with a spectral range of 300–700 nm [19]. The disclosed structure exhibited a reduction in both absorption bandwidth and absorption level. However, the maximum absorption value was enhanced to 98%. Moreover, previous references [17,18,19] did not ascertain the insensitivity to polarisation or demonstrate stability in the presence of oblique incident angles. A different study [20] proposed developing an absorber with Au and Si components, demonstrating enhanced stability at incident angles of up to 65°. Notwithstanding, the recorded absorption level exhibited a reduction of 20%. Meanwhile, another study proposed an absorber predominantly composed of nickel (Ni) and (Si) materials, exhibiting a high absorption coefficient exceeding 90% over a range of wavelengths between 400 and 700 nm [21]. Upon an examination of the previous studies, it was evident that achieving polarisation insensitivity and oblique incident angle stability with a high-absorption-level MMA across the wavelength spectrum of 380–750 nm is a crucial requirement for optical metamaterial absorber applications in the visible region. Based on the findings, the study indicated a slight degradation in stability at a 60° incident angle, while no substantial impact on the sensitivity towards polarisation was reported. The present study demonstrated a wide-band absorption that was yielded within the spectral regions of 539.2–581.20 THz and 644.4–661.60 THz, exhibiting an absorption rate exceeding 90%. The average wide-band absorption performance of the PMA was observed to be 99.992%.

## 2. Materials and Design Technique

The proposed design for the absorber on a nanoscale adopted a MIM architecture. Nickel (Ni-lossy metal) was chosen as the front resonator and ground slab. The rationale for selecting Ni as the metallic component of the structure was attributed to its properties. Ni exhibits exceptional resistance to elevated temperatures and corrosion while boasting a comparatively low cost and ease of manufacturability [22]. The autologous behaviour exhibited by Ni, along with its significantly high melting temperature of approximately 1453 °C and low emissivity at longer wavelengths contribute considerably to the successful attainment of desired outcomes in the design process [23]. Meanwhile, the rationale for selecting the dielectric insulator (SiO_2_-optical) as an insulating spacer was attributable to its lossless characteristics within the intended wavelengths [24]. The exceptional stability under elevated temperatures of SiO_2_ can be attributed to its significantly elevated melting point, which is approximately 1600 °C. According to the literature, SiO_2_ presents a relatively lower permittivity than its non-real dielectric constant component when observed within the visible spectrum [25]. Consequently, it can be observed that during a state of breakdown, there is a reduction in the real component of permittivity, which leads to the emergence of a more confined propagating wave, in accordance with the evanescent wave characteristic. The present study shows that the anisotropic tendency observed can provide a notable contribution towards the polarisation and propagation regulation that occur within a given substrate. This contribution arises from the fact that the birefringence characteristics of the dispersion relation pairs effectively correlate with the refractive index of SiO_2_. Moreover, the selected dielectric material must maintain optimal inductance and coupling capacitance levels. The proposed configuration exhibited remarkable heat resistance owing to the constituents’ exceptionally high melting points.

The design methodology achieves high accuracy in the physical dimensions of the unit cell by implementing a symmetric structure to ensure that close-to-unity absorption is attained while mitigating the effects of insensitive polarisation [26]. The octagonal disk nanoparticle is characterised by a regular spatial arrangement of a resonator measuring 560 nm along the *x*- and *y*-axis. This arrangement is denoted by its graphic representation in green. Table 1 presents the additional dimensions of the octagonal nanoantenna, which include tr = 65 nm, tg = 75 nm, hs = 250 nm, ws = ls = 560 nm, and Rr = 220 nm. The thickness of the underlying material substrate is measured at 250 nm, depicted in yellow. The thickness of the background Ni layer is significant as it serves as a photosensitive reflector, effectively reducing the level of transmittance to near-zero values. The present study highlights the coupling interaction between the Ni layer underlying the resonator and the topmost Ni layer, as depicted in Figure 1. Figure 1 consists of (a) a schematic diagram of PMA with a perspective view, (b) bottom view, (c) conceptual layout, and (d) numerical analysis setup. The structural dimensions of the specimen are denoted as 560 × 560 × 250 nm^3^. The ultrathin composition exhibited aptness for employment within solar thermophotovoltaic (STPV) cells.

## 3. Results and Discussion

This section discusses the absorption characteristics, metamaterial characteristics, polarisation insensitivity, and incident angle stability of the design structure for solar energy harvesting.

### 3.1. Absorption Characteristics

The absorption characteristics of the PMA structure are defined by two fundamental parameters, namely reflectance (*R*(ω)) and transmittance (*T*(ω)), which are quantities that vary with frequency. The expression denoting PMA is formulated as *A*(ω) = 1 *− R*(ω) *− T*(ω), where *R*(ω) and *T*(ω) represent the coefficients of reflection and transmission, respectively. The rates of the reflection and transmission of the absorber are contingent upon the scattering parameters, namely $R\left(\omega \right)=S_{11}{}^{2}$ and $T\left(\omega \right)=S_{21}{}^{2}$. Accordingly, optimising the geometric structure of PMA to attain resonance frequency, whereby $Z\left(\omega \right)=Z_{0}$, is deemed essential. Yielding this frequency matches the impedance with that of free space impedance. The absorption properties of a given structure are contingent upon its impedance-matching capabilities. The relative impedance (*Z*) of the three-layer sandwich model under consideration was computed utilising Equation (1) in [26].

Moreover, Figure 2 depicts the relative impedance of transverse electric (TE) and transverse magnetic (TM) modes. When the real part of the impedance closely approximates unity, and the imaginary component approaches null values, the effective impedance of the structure is in accordance with the impedance of free space, facilitating high absorption levels.
(1)Z=[{(1+S11)2−S212}/{(1−S11)2+S212}]=(με)/Z0=(μrεr)

In the context of perfect metamaterial absorption, the permeability (designated as µ) and permittivity (defined as ε) can be expressed as $\mu =\mu_{r}\mu_{0}$ and $\varepsilon =\varepsilon_{r}\varepsilon_{0}$, respectively. The relative permeability and permittivity of the medium are represented by $\mu_{r}$ and $\varepsilon_{r}$, respectively. The parameters known as permeability and permittivity of free space are denoted by $\mu_{0}$ and $\varepsilon_{0}$, respectively. The impedance of free space is equivalent to the quotient of the permeability of the vacuum, $\mu_{0}$ and the permittivity of the vacuum, $\varepsilon_{0}$, as denoted by the Equation $Z_{0}=\mu_{0}/\varepsilon_{0}$. The reflectance (*R*) of the metamaterial for the TE and TM modes can be obtained through Equations (2) and (3), respectively [27].
(2)$$R_{TE}={\left|r_{TE}\right|}^{2}=\left|\left(\mu_{r}\;\cos\;\theta -\sqrt{n^{2}-\sin\theta }\;\right)/\left(\mu_{r}\;\cos\;\theta +\sqrt{n^{2}-\sin\theta }\;\right)\right|$$
(3)$$R_{TM}={\left|r_{TM}\right|}^{2}=\left|\left(\varepsilon_{r\;}\cos\;\theta -\sqrt{n^{2}-\sin\theta }\;\right)/\left(\varepsilon_{r}\;\cos\;\theta +\sqrt{n^{2}-\sin\theta }\;\right)\right|$$

The equations consist of *n*, signifying the refractive index, and *θ*, representing the wave’s angle of incidence during a distinctive incidence. Equations (2) and (3) are modified to yield the following expressions:(4)$$R_{TE,\;TM}={\left|\left(Z-Z_{0}\right)/\left(Z+Z_{0}\right)\right|}^{2}={\left|\left(\sqrt{\mu_{r}}-\sqrt{\varepsilon_{r}}\right)/\left(\sqrt{\mu_{r}}+\sqrt{\varepsilon_{r}}\right)\right|}^{2}$$

Equation (4) delineates the reflectance of the MMA and is primarily influenced by the confluence of impedance matching and metamaterial properties. This study conducted a numerical investigation of a given structure using the FIT method-based CST EM simulator. The boundary condition configuration includes a unit cell boundary condition imposed in the *x* and *y* directions, while an open boundary condition is applied in the *z*-direction. The EM wave’s propagation direction is aligned with the *z*-axis, while the electric and magnetic fields are parallel to the *x*- and *y*-axis. Figure 1 displays the proposed structure’s diagrammatic representation, the conceptual configuration featuring appropriate dimensions, and the simulation setup. During analysis, the numerical simulations employ a frequency domain solver to evaluate the structure’s performance. Hence, this study aims to elucidate the reflection and absorption parameters in conjunction with the absorption of the design structure via varying the thicknesses of Ni metals and SiO_2_-optical materials. The effects of the resonator’s shape, the radius of the resonator, and the polarisation effect based on TE and TM were also investigated.

Figure 3 illustrates the reflectance and absorption properties of the TE and TM modes. Examining the polarisation of solar energy in the TE and TM modes within the numerical analysis is commonly conducted using the periodic boundary condition. As depicted in Figure 3, the proposed PMA exhibited autonomy in TE and TM polarisation attributed to the symmetrical design structure of the device. The absorption peaks generated by the TM mode at resonance frequencies of 549.20 THz and 653.20 THz demonstrated absorption values of 99.987% and 99.997%, respectively. Conversely, the TE mode exhibited maximum absorption peaks at the resonance frequencies of 549.20 THz and 652.40 THz with individual values of 99.979% and 99.993%, respectively. Therefore, due to its near-perfect absorption properties, the proposed PMA can harvest solar energy in the optical spectrum.

### 3.2. Metamaterial Characteristics

The metamaterial properties of the suggested PMA for TE and TM modes are displayed in Figure 4a and 4b, respectively. Figure 4a–c in the TE mode displayed negative permittivity, permeability, and refractive index at 400–405.60 THz, 408.40–542.80 THz, 550.80–648.40 THz, and 660–668.80 THz; 400–400.80 THz, 413.20–546 THz, and 552.40–550.80 THz; 400–404.80 THz; 410–544.80 THz; and 551.20–650.40 THz. Whereas, in the TM mode, the permittivity, permeability, and refractive index were negative at 400–405.60 THz, 409.20–649.2.80 THz, 654–687.20 THz, and 703.20–728 THz; 400–401.20 THz, 412.80–651.60 THz, and 654.40–695.20 THz; and 400–404.80 THz; 410.4–544.80 THz, 551.20–650.80 THz, and 654–691.60 THz, respectively. Consequently, these negative values indicated that the proposed MMA exhibited high-level wide-band absorption properties in the TM mode. On the other hand, the TE mode also achieved negative values for the metamaterial property.

### 3.3. Polarisation Insensitivity and Incident Angle Stability

The observed polarisation insensitivity of the proposed PMA provided empirical validation for its efficacy in absorbing incident radiation. The solar absorber, utilising a PMA structure, exhibited individualistic properties in polarisation and frequency, resulting in an absorption efficiency exceeding 90% within the frequency ranges of 539.2–581.20 THz and 644.40–661.60 THz. To appraise the absorption features of the design for its interaction with EM waves, the characteristic response was analysed at varying polarisation angles of the incident EM rays, as depicted in Figure 5. Figure 5 illustrates the absorption characteristics of diverse incident polarisation angles (*φ*) for TE and TM modes. The direction of wave propagation is aligned with the TE mode to the *z*-axis. The components of the electric field vector, denoted as Ex, and the magnetic field vector, denoted as Hy, align along the *x*- and *y*-axis, respectively. In the TM mode, the direction of wave propagation was aligned along the *z*-axis. Meanwhile the Hx (magnetic) and Ey (electric) field vectors also aligned along the *x*- and *y*-axis, respectively. The proposed PMA attained distinctive absorption characteristics in the face of polarisation incident angles (*φ*) up to 90° in regard to its axial and rotational symmetry strength.

Compliance with the design framework on varying polarisations of EM radiation from the solar source assumes a significant role in the research on PMA architecture for photovoltaic implementation. Designing a solar cell that effectively captures solar energy has proven challenging, as it requires an apparatus capable of containing EM radiation of an arbitrary polarisation. Figure 5 illustrates the absorption rate for different polarisation angles (*φ*), including the TE and TM modes. The aforementioned absorption rate was examined through numerical analyses at distinct angles ranging from 0° to 90° with a 30° interval. Based on the numerical analysis, the highest absorption peaks were yielded at 99.993%, 99.984%, 99.987%, and 99.989% for polarisation angles of 0°, 30°, 60°, and 90° at 652.40 THz in the TE mode. According to Table 2, the highest peaks of absorption, reaching values of 99.997%, 99.922%, 99.941%, and 99.921%, were also observed in the TM mode at a frequency of 653.2 THz for four distinct polarisation angles, namely 0°, 30°, 60°, and 90°. As depicted in Figure 5a,b, the proposed design configuration exhibited a propensity for nearly constant and optimal absorption in response to all polarisation angles of the incoming EM wave. The absorption response of the proposed structure is depicted in Figure 5a,b, revealing that the suggested structure’s angular disposition led to different treatments of the two modes of EM waves, namely TE and TM. Therefore, the proposed PMA exhibited favourable potential for elucidating the operational mechanism of dynamic absorber devices that operate within distinct spectral ranges of solar irradiance. Furthermore, the material’s insensitivity to TE and TM waves rendered it a promising candidate for the photovoltaic collection of solar energy applications.

Figure 6a,b depict the alterations in the angle of incidence, ranging from 0° to 60°, for TE and TM modes. As the incidence angles increased, a discernible decrease in the average absorption was observed, accompanied by a shift towards higher frequencies. The correlation between higher incidence angles and longer path lengths reduced the coupling impact. The EM dipolar resonance of the structure experienced a decline due to the decrease in the coupling effect, which subsequently diminished the wave-confining proficiency of the dielectric layer. The determination of absorption was performed using specular reflection. This approach is justified by the simulation outcomes, which revealed a deviation of 15° from the normal incidence, reducing the reflected field strength, coupled with a shift towards higher frequencies. This phenomenon strongly suggested that most reflected energy was concentrated in this direction. Based on the numerical analysis, the TE mode yielded the highest peaks of absorption at 99.979%, 99.992%, 99.922%, 99.534%, and 99.539%, correspondingly, for incident angles of 0°, 15°, 30°, 45°, and 60°, at frequencies of 549.20 THz, 548.80 THz, 514.40 THz, 498.00 THz, and 485.20 THz, respectively. Meanwhile, for the TM mode, the maximum peaks of absorption of 99.987%, 99.994%, 99.857%, 99.928%, and 99.489% were recorded at corresponding frequencies of 549.20 THz, 549.20 THz, 514.40 THz, 496.40 THz, and 485.20 THz, across five distinct incident angles (0°, 15°, 30°, 45°, and 60°). Table 3 tabulates the readings from the TE and TM modes.

## 4. Parametric Sweep

This section discusses the effects of the absorption properties of the crucial geometric parameter variation.

### 4.1. Sweep of Top, Ground Resonator, and Substrate Thickness

The relationship between the absorption and geometric parameters of the proposed PMA was investigated in depth in this study. The thicknesses of the top (tr) and bottom (tg) metals were altered to assess the response of the PMA. The parameter tr was varied between 55 and 95 nm in increments of 10 nm, as illustrated in Figure 7a. At 653.20 THz and 549.20 THz, the maximum absorption values were estimated at 99.997% and 99.987%, respectively. The highest absorption was achieved at an optimal tr value of 65 nm due to the perfect impedance matching between the free space, the metamaterial unit cell, and the occurrence of resonance conditions.

The impedance match was significantly impacted by the capacitance engendered by the metal plane and metal resonator. The excellent metallic property of Ni and its susceptibility to structural ruptures induced by EM waves enabled the metal resonator to establish effective capacitance with the back metal plane. This capacitance is inversely proportional to the thickness of the resonator. Consequently, a minor alteration in the resonant wavelength was observed. Meanwhile, the parameter denoted by tg exhibited a range of variability spanning 25 to 225 nm, with incremental steps of 50 nm. The observation is graphically represented in Figure 7b.

Conversely, variations in the thickness of the top and bottom metal layers corresponded with the diminishing absorption peaks. A systematic exploration was required to comprehend the PMA’s reaction under varying substrate thicknesses (ts). Based on Figure 7c, parameter ts was modified within a range from 175 to 275 nm, with incremental adjustments of 25 nm. Simultaneously, the absorption peaks were measured to be 99.997% and 99.987% at frequencies of 653.2 THz and 549.20 THz, respectively. Maximum absorption was attained at the proposed spectral position of 250 nm due to the favourable matching of the free space impedance and the metamaterial unit cell, along with a resonance condition. As the width of the dielectric increased, the resonant frequency shifted. Moreover, the capacitive and inductive modification effects can be achieved by manipulating the dielectric thickness, leading to substantial alterations in peak absorption values.

### 4.2. Sweep of Resonator’s Material and Resonator Shape

The absorption values are subject to change, owing to the materials employed in the resonator. The conducting materials, such as aluminium, gold, silver, tin, and nickel, were also considered for the analysis of the absorptive properties of the design structure (Figure 8a). The maximum absorption peaks were 99.312%, 98.784%, 97.589%, 98.152%, and 99.987%; 98.573%, 97.294%, 95.148%, 97.139%, and 99.997% for the first and second resonance at 549.20 THz and 653.2 THz, respectively, for different metal resonators (aluminium, gold, silver, tin, and nickel). The absorption was measured to be 99.997% and 99.987% at 653.2 THz and 549.20 THz concurrently for the Ni metal resonator. The proposed design structure with a Ni resonator yielded maximum absorption. The resonant frequency was shifted due to the sweep of the resonator material. Therefore, altering the resonator material can change the materials’ properties. Consequently, the capacitive and inductive effects and varying peak absorptions are also changed. Different resonator shapes such as square, pentagon, hexagon, octagon, and circle, with separate effective areas but the same resonator radius, were created and investigated separately to determine the highest absorption peak. Figure 8b describes the compared and examined impacts with the suggested structure. The most extreme absorption peaks were 83.982%, 96.175%, 99.227%, 99.987%, and 90.198%; 89.159%, 91.706%, 99.925%, 99.997%, and 96.188% for the first and second resonance at 549.20 THz and 653.20 THz, respectively, for different shapes (square, pentagon, hexagon, octagon, and circular). As for the octagonal metal resonator, the absorption was recorded to be 99.997% and 99.987% at 653.2 THz and 549.20 THz, respectively. In short, an octagonal shape is more suitable for solar energy harvesting for high absorption peaks.

The reaction of resonators with various shapes, including square, pentagon, hexagon, octagon, and circular configurations, were investigated, where each resonator possessed the same effective area of 136,894.72 nm but differed in radius. The respective properties of the resonators were analysed independently. Figure 8c illustrates a comparative analysis of the observed impacts with those derived from the suggested structure. The first and second resonances occurred at 549.20 THz and 653.20 THz for shapes with identical practical areas, including square, pentagon, hexagon, octagon, and circle. Maximum absorption peaks of 86.324%, 99.510%, 99.816%, 99.987%, and 99.971%; 87.356%, 99.788%, 99.990%, 99.997%, and 99.881% were obtained for various shapes (square, pentagon, hexagon, octagon, and circular) at 549.20 THz and 653.20 THz, respectively. The octagonal metal resonator, with an identical effective area, exhibited absorption values of 99.997% and 99.987% at 653.20 THz and 549.20 THz, respectively. Consequently, the utilisation of octagonal morphology would be more appropriate for the extraction of solar energy, given its capacity for generating elevated levels of absorption peaks.

### 4.3. Response of Electric Field and Magnetic Field

The electric field (V/m) and magnetic field (A/m) distributions of the PMA designed structure for a normal incident angle of the TE and TM modes, respectively, are presented in Figure 9 and Figure 10.

The absorption characteristic can be examined from the EM phenomenon. The characteristic of an EM field can be heightened and disturbed at various locations of the absorber. The E-field and H-field characteristics of the designed absorber at resonance frequencies of 549.20 THz and 652.40 THz were demonstrated in this study. The EM field characteristics are also meaningfully influenced by the metamaterial properties of the absorber. From Equations (5) and (6), the relationship between the characteristic of EM and the metamaterial properties can be realised.
(5)$$D=\varepsilon_{eff}\varepsilon_{0}E$$
(6)$$B=\mu_{eff}\mu_{0}H$$
where *D* = the electric flux density, $\varepsilon_{eff}=$ effective permittivity, $\varepsilon_{0}$ = is free space permittivity, *B* = the magnetic flux density, $\mu_{eff}$ = effective permeability, $\varepsilon_{0}$ = is free space permeability, the electric and magnetic field intensities are *E* and *H*, respectively.

The maximum e-field is integrated into the metal area of the resonator, where a maximum intensity of the h-field appeared in the leaping part of the resonator at a 652.40 THz resonance frequency in the TE mode. The ultimate absorption peak of 99.993% was attained at 652.40 THz, with an e-field intensity of more than 549.20 THz. However, a high h-field was created in the dielectric substrate with a perspective view of 652.4 THz resonance frequency in Figure 9, leading to an enlarged absorption peak. Figure 10 illustrates the EM performance of the TM mode, where the field distribution is similar to that of the TE mode but interchanges from vertical to horizontal in the TM mode.

As depicted in Figure 11, the suggested structure exhibited a frequency range within the visible spectrum. Figure 11 is the numeric absorption spectrum of the proposed PMA, as determined using the finite element method. The data collected from Figure 11 indicate that the two maximal absorption peaks at 99.940% and 99.523% are positioned at 504.40 THz and 670 THz, respectively. The variances in the numerical methodology, software model, and substrate material properties resulted in absorption peaks of comparable magnitude with slight alterations in the resonance frequencies.

Table 4 compares the proposed PMAs and the published PMAs. The present discourse carefully examined various aspects of the parameters associated with PMA, including, but not limited to, references, dimensions, materials, bandwidth, polarisation insensitivity, incident angle stability, average absorption, and absorption peak. In this study, the PMA of a compact octagonal form was scrutinised and yielded maximal absorption levels, specifically at resonance frequencies of 549.20 THz and 653.20 THz, exhibiting absorption perfection of 99.987% and 99.997%, respectively. The PMA demonstrated a noteworthy attribute of polarisation insensitivity and incident angle stability. The proposed photovoltaic material exhibited a substantial increase in absorption relative to the cited works, viable for the visible spectrum regarding energy conversion.

## 5. Conclusions

In the present study, we performed a numerical analysis to investigate the appropriateness of the proposed tri-layered design structure (Ni–SiO_2_–Ni) for its deployment as a solar energy harvesting device. The proposed assessment of the system and the geometrical parameters were analysed to achieve nearly perfect absorption. This claim was corroborated through the use of HFSS numerical analysis. The concurrent demonstration of the absorption calculation of the TE and TM modes exhibited a homogeneous absorption profile, which concomitantly enhanced the appropriateness of the predicted PMA. A maximum absorption value of 99.997% was achieved at a frequency of 653.20 THz, while an average wide-band absorption performance of the PMA was observed at 99.992%. Both the TE and TM modes exhibited polarisation insensitivity and incident angle stability. The characteristics mentioned above render the predicted PMA a promising option for a range of potential uses, including, but not limited to, solar radiation absorption and light confinement. Ultimately, a comprehensive analysis of the published PMAs elucidated the distinct characteristics and advantages of the proposed PMA.

## Figures and Tables

**Figure 1 nanomaterials-13-01882-f001:**
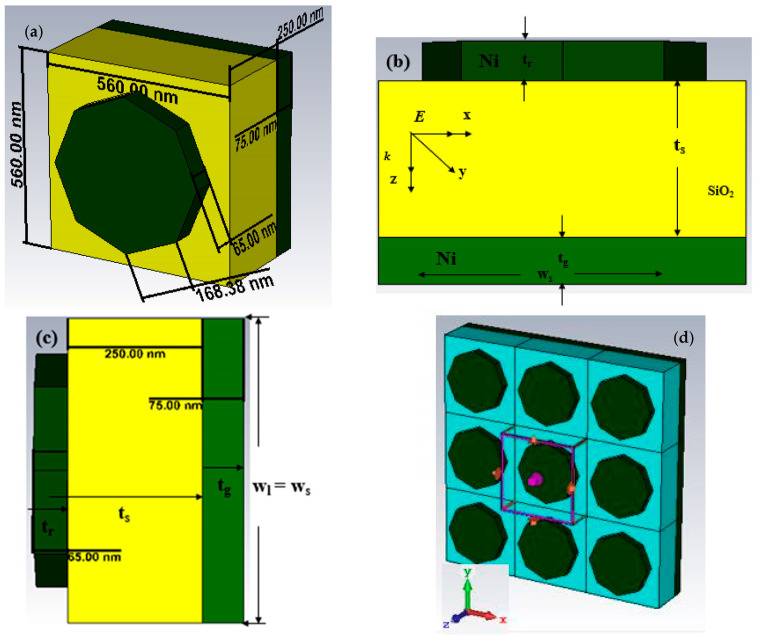
Schematic diagram of PMA with (**a**) perspective view, (**b**) bottom view, (**c**) conceptual layout, and (**d**) numerical analysis setup.

**Figure 2 nanomaterials-13-01882-f002:**
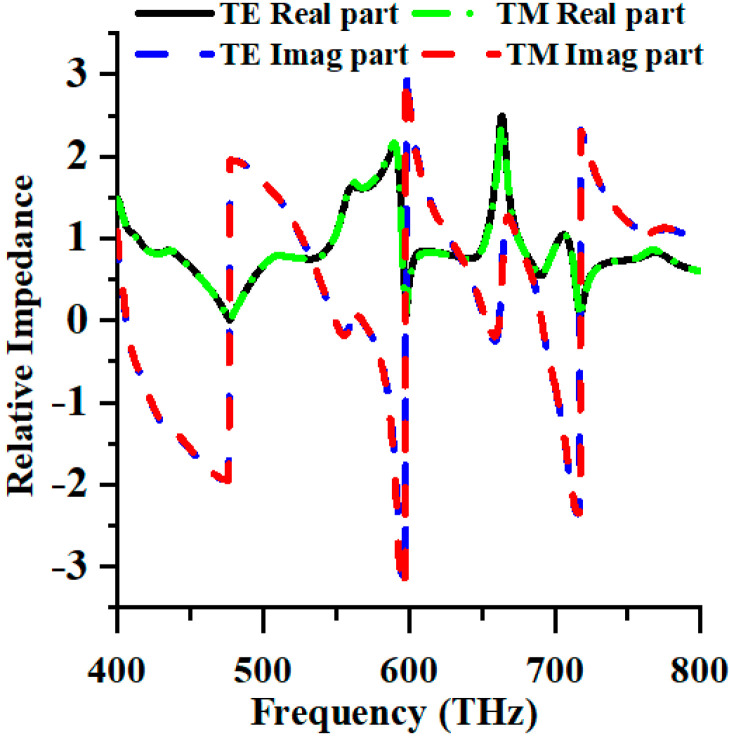
The relative impedance of PMA.

**Figure 3 nanomaterials-13-01882-f003:**
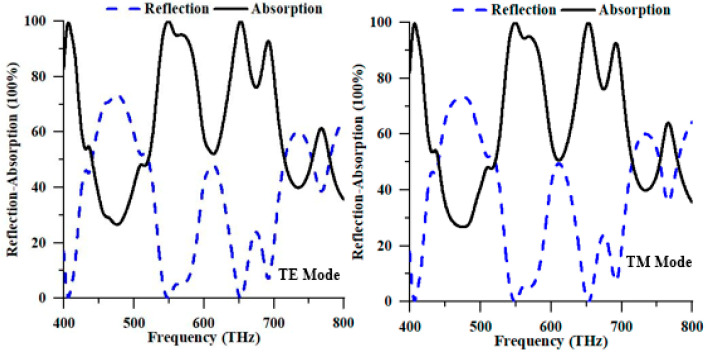
Numerical reflection and absorption spectrum of the proposed structure in TE and TM modes.

**Figure 4 nanomaterials-13-01882-f004:**
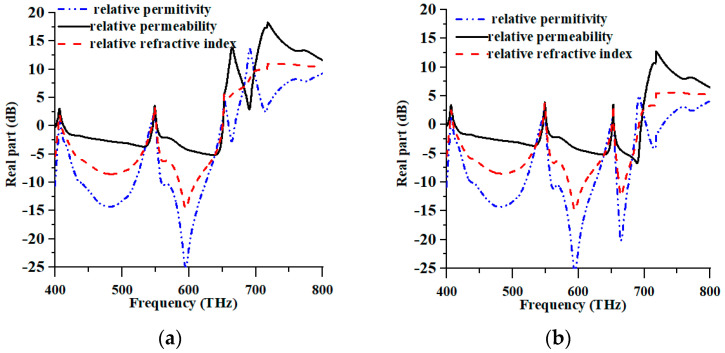
Metamaterial properties of (**a**) TE and (**b**) TM modes.

**Figure 5 nanomaterials-13-01882-f005:**
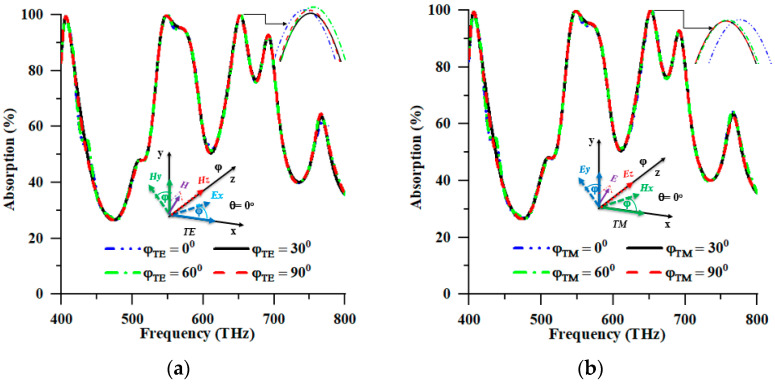
Polarisation insensitivity of (**a**) TE and (**b**) TM modes.

**Figure 6 nanomaterials-13-01882-f006:**
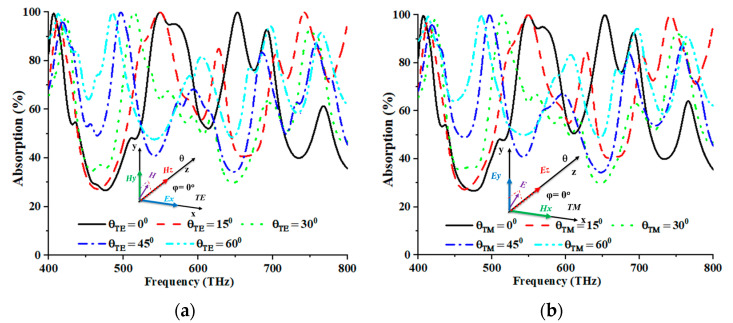
Incident angle stability of (**a**) TE and (**b**) TM modes.

**Figure 7 nanomaterials-13-01882-f007:**
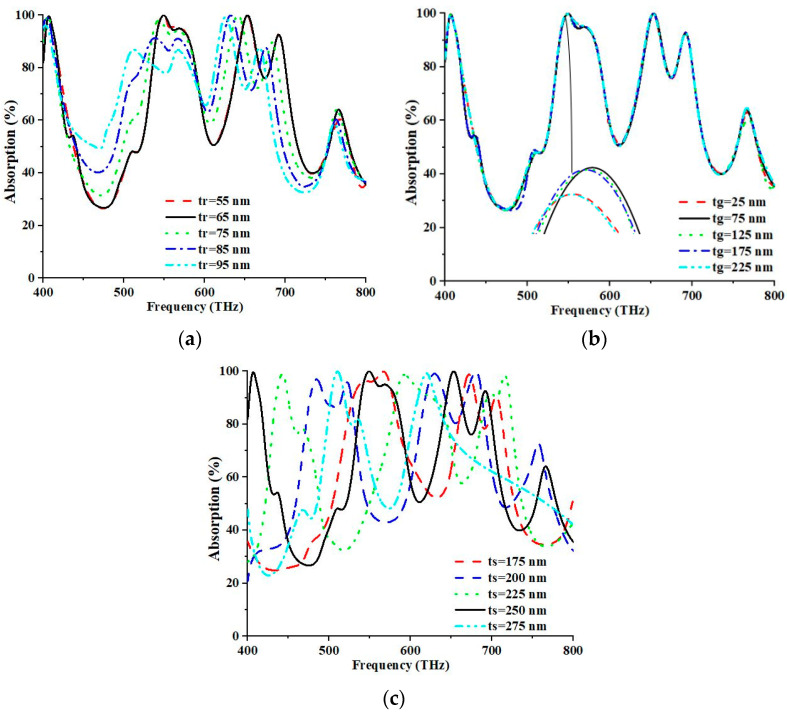
(**a**) Parametric analysis for parameter thickness of (**a**) the top, (**b**) ground resonator, and (**c**) the substrate.

**Figure 8 nanomaterials-13-01882-f008:**
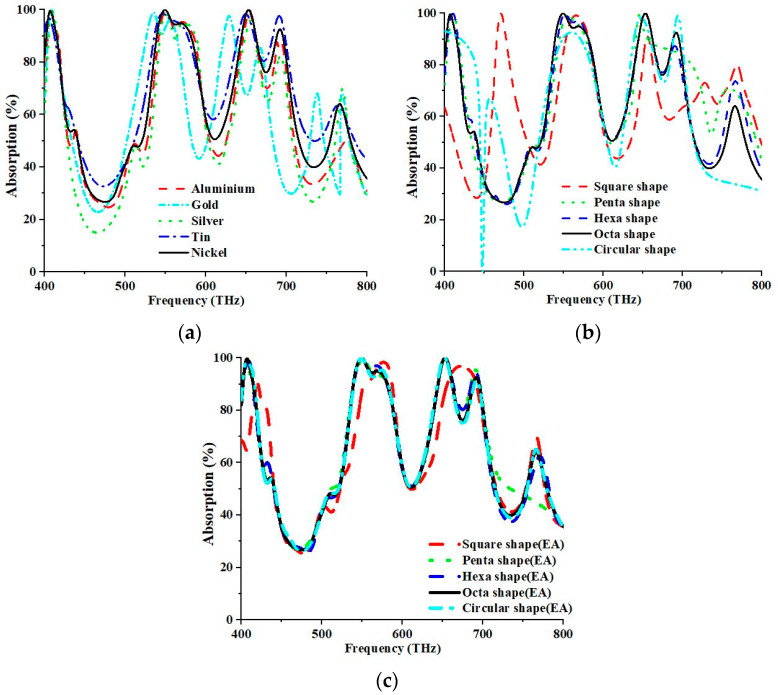
Parametric analysis for parameters of (**a**) various metal resonators, (**b**) different resonator shapes, and (**c**) resonators’ shape with the same effective area.

**Figure 9 nanomaterials-13-01882-f009:**
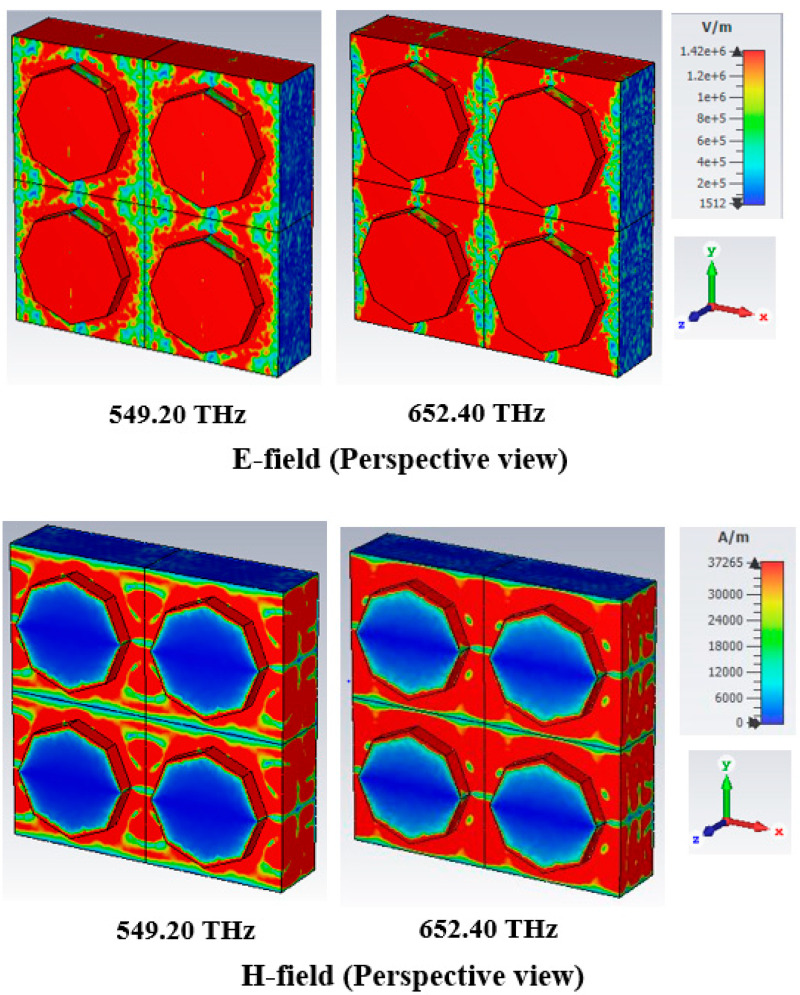
E-field and H-field distributions at TE mode.

**Figure 10 nanomaterials-13-01882-f010:**
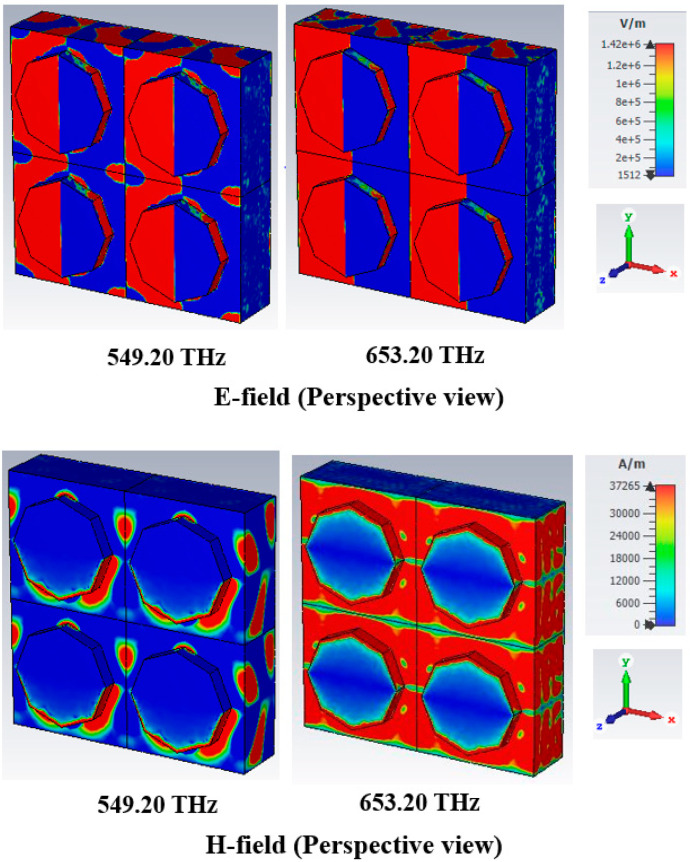
E-field and H-field distributions at TM mode.

**Figure 11 nanomaterials-13-01882-f011:**
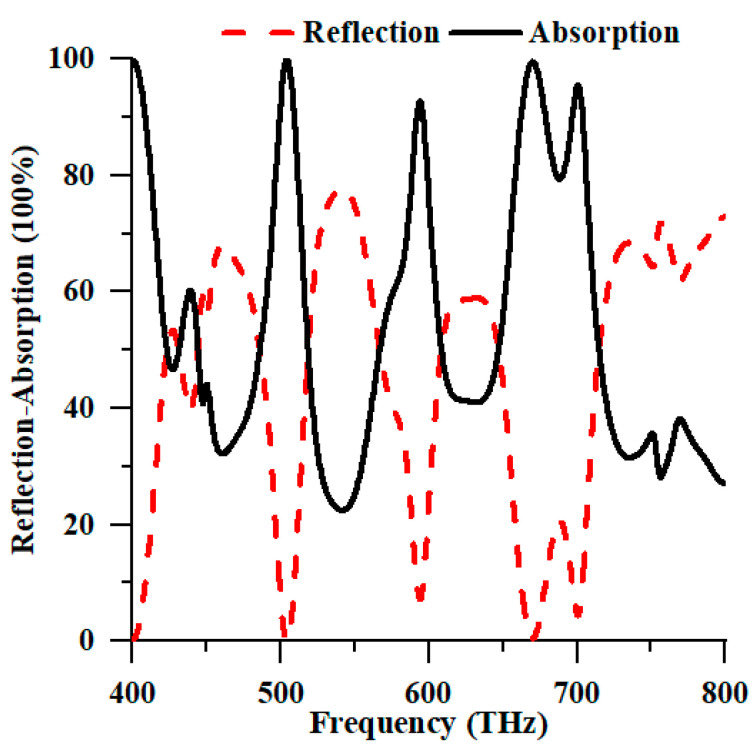
Numerical reflection and absorption spectrum of the suggested structure in HFSS.

**Table 1 nanomaterials-13-01882-t001:** Parameters of the suggested PMA structure.

Parameters	Value (nm)	Parameters	Value (nm)
Substrate width, ws	560	Resonator’s sidearm, s	168.38
Substrate length, ls	560	Resonator’s thickness, tr	65
Substrate height, hs	250	Ground plane’s thickness, tg	75
Resonator’s radius, Rr	220		

**Table 2 nanomaterials-13-01882-t002:** Absorption properties of different polarisation angles (*φ*).

Mode	Polarisation Angle (*φ*)	Absorption Peak at 549.20 THz	Absorption Peak	Frequency
TE	0°	99.979%	99.993%	652.40 THz
	30°	99.667%	99.984%	
	60°	99.966%	99.987%	
	90°	99.646%	99.989%	
TM	0°	99.987%	99.997%	653.20 THz
	30°	99.676%	99.922%	
	60°	99.966%	99.941%	
	90°	99.646%	99.921%	

**Table 3 nanomaterials-13-01882-t003:** Absorption properties of different incident angles (*θ*).

Mode	Incident Angle (*θ*)	Absorption Peak	Frequency at THz	Absorption Peak	Frequency at THz
TE	0°	99.979%	549.20	99.993%	652.40
	15°	99.992%	548.80	99.939%	742.00
	30°	99.922%	514.40	92.292%	748.80
	45°	99.534%	498.00	83.467%	685.60
	60°	99.539%	485.20	91.479%	764.00
TM	0°	99.987%	549.20	99.997%	653.20
	15°	99.994%	549.20	84.766%	627.60
	30°	99.857%	514.40	92.179%	751.20
	45°	99.928%	496.40	83.633%	686.00
	60°	99.489%	485.20	94.232%	696.80

**Table 4 nanomaterials-13-01882-t004:** Performance comparison between the proposed PMAs with published PMAs.

Ref.	Dimension (nm^3^)	Materials	Bandwidth (nm)	Polarisation Insensitivity (*φ*) and Incident Stability (*θ*)	Average Absorption	Peak Absorption
[28]	500 × 500 × 450	W, SiO_2_, Au	1759.80	N/A	93.17%	98.63%
[29]	1000 × 1000 × 310	Au, SiO_2_	1234	N/A	80.24%	96.40%
[30]	400 × 400 × 295	Ti, SiO_2_	1376	Yes, θ≤60°	94.60%	97.7%
[31]	1000 × 1000 × 320	Ti, Al_2_O_3_, W	1300	Yes, θ≤60°	94.00%	99.99%
[32]	600 × 600 × 550	Au, SiO_2_, Au	1500	No, θ≤40°	98.53	99.60%
[33]	800 × 800 × 300	Cu, SiO_2_, Cu	700	No, θ≤60°	95.53	99.90%
[34]	1000 × 1000 × 530	Au, SiO_2_, Au	350	No, θ≤60°	97.30%	99.90%
[35]	500 × 500 × 130	Ni, SiO_2_, Ni	370	Yes, No	95.76%	99.93%
Proposed	560 × 560 × 75	Ni, SiO_2_, Ni	375	Yes, θ≤60°	99.992%	99.997%

## Data Availability

All the data are available within the manuscript.

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
