# Peer review of "An Innovative Polarisation-Insensitive Perfect Metamaterial Absorber with an Octagonal-Shaped Resonator for Energy Harvesting at Visible Spectra"

_nanomaterials, 2023, doi:10.3390/nano13121882_

Round 1

Reviewer 1 Report

The Perfect Metamaterial Absorber (PMA) is an attractive absorber of optical wavelengths, with potential for solar energy and photovoltaic applications. Using perfect metamaterials as a solar cell can improve efficiency by amplifying incident solar waves on the PMA. This manuscript proposes an ultrawide-band octagonal PMA for the visible wavelength spectrum, featuring high absorbance peaks in TE and TM modes and insensitivity to polarization angle. The work provides a clear exposition of the physical mechanisms. But before it can be accepted by the journal, the following concerns are suggested to be solved.

1.     Please highlight the innovative nature of the work.

2.     Please analyze the electromagnetic response of the structure at different incidence angles.

3.     For comparison with other related works, the relevant parameters listed in Table 5. However, some of the compared literature is long time ago, please compare more with the related work in the last three years. In addition to the peak absorbance, the average absorbance and bandwidth are also important, and these two parameters should be compared.

4.     Some related publications may help enrich the introduction and are suggested to be mentioned in this background.

A refractory metamaterial absorber for ultra-broadband, omnidirectional and polarization-independent absorption in the UV-NIR spectrum, Nanoscale, 2018, 10(17): 8298-8303.

A novel double 3D continuous phase composite with ultra-broadband wave absorption from gigahertz to UV–vis-NIR for extremely cold environment, Chemical Engineering Journal, 2022, 436: 135220.

Catenary Electromagnetics for Ultra-Broadband Lightweight Absorbers and Large-Scale Flat Antennas, Advanced Science 6: 1801691 (2019)

The English expression is suggested to be further polished.

Author Response

As attached.

Reviewer 2 Report

The authors present a broadband metamaterial perfect absorber based on a conventional metal-insulator-metal configuration. A nickel octagonal disk act as the top optical resonator. Two separate absorption bands are observed in the visible wavelength range. While systematic studies are performed, considerable improvement is required for this work. My comments are:

1. Absorption and absorbance are two different concepts and the authors need to double check on this. It is better to use the one that has similar definition as reflection. In Fig.3, at some frequencies, reflection+aborbance is larger than 1. Please check this.

2. In Fig.8 and Fig.9, the near field profile is very hard to perceive. Please modify the field vector and/or color bar to make it more clearly presented. 

3. For section "5. Performance analysis of array unit cell and unit cell", I do not understand its significance to this work. Is all the analysis before this section based on a single cell or a periodic array? If it is based on an array, then section 5 is unnecessary unless disks with different sizes are included in order to create more absorption bands. Fig. 10 -15 also show that the spectra exhibit little changes confirming that such modifications do not affect the spectra very much. 

4. The writing also needs to be improved to clearly and concisely present the results and corresponding analysis. Redundant description can be found in the manuscript, Such as line 279-284. Line 222 and 227 just repeat each other.

5. The references need to be better organized. In the Introduction, the authors mentioned the work by Landy et al with no reference. They also mentioned Li et al with no reference. Additionally, some recent development in this field should be included, such as Optics Express 29 (17), 27084-27091, 2021, as metamaterial perfect absorbers have already been achieved in various configurations for different applications. 

See the comments above.

Author Response

As attached.

Round 2

Reviewer 1 Report

I'm glad that the revised version has solved the reviewers' concerns and the manuscript quality has been greatly improved. I suggest it can be accepted now.

Reviewer 2 Report

I think the authors have addressed all my concerns.